# Visual Perception of Facial Emotional Expressions during Saccades

**DOI:** 10.3390/bs9120131

**Published:** 2019-11-27

**Authors:** Vladimir A. Barabanschikov, Ivan Y. Zherdev

**Affiliations:** Institute of Experimental Psychology, Moscow State University of Psychology and Education, 29 Sretenka street, Moscow 127051, Russia; vladimir.barabanschikov@gmail.com

**Keywords:** transsaccadic processing, facial expression, emotional perception, saccadic suppression, gaze contingency, visual recognition

## Abstract

The regularities of visual perception of both complex and ecologically valid objects during extremely short photo expositions are studied. Images of a person experiencing basic emotions were displayed for as low as 14 ms amidst a saccade spanning 10 degrees of visual angle. The observers had a main task to recognize the emotion depicted, and a secondary task to point at the perceived location of the photo on the screen. It is shown that probability of correct recognition of emotion is above chance (0.62), and that it depends on its type. False localizations of stimuli and their compression in the direction of the saccade were also observed. According to the acquired data, complex environmentally valid objects are perceived differently during saccades in comparison to isolated dots, lines or gratings. The rhythmic structure of oculomotor activity (fixation–saccade–fixation) does not violate the continuity of the visual processing. The perceptual genesis of facial expressions does not take place only during gaze fixation, but also during peak speed of rapid eye movements both at the center and in closest proximity of the visual acuity area.

## 1. Introduction

According to common notion, visual processing is performed discretely at the very moments of stable gaze fixations (average duration equals 250 to 350 ms). During rapid (saccadic) eye movements (average duration equals 30 to 60 ms), visual sensation decreases dramatically, making it difficult to perceive new visual data [1,2,3]. The scale of saccadic inhibition effect varies depending on the saccade amplitude, the brightness and contrast of the stimulus, and position of its projection onto the subject’s visual field. Texture detalization of the stimulus may exaggerate inhibition effect, while the perceived localization of the stimulus may not correspond with the actual one, and the surrounding space dimensions may appear somewhat compressed, or narrowed [4,5,6,7,8,9]. The analysis of perceptive ability during saccades allows us to reveal the nature of visual processing (discretization vs continuity), patterns of perceptiogenesis in regards to complex stimuli, conditions for stable perception of visible movement direction, and more.

The effects of saccadic inhibition have been experimentally studied by means of relatively simple test stimuli exposure [2] to naïve subjects during detection and identification tasks (light dots, simple geometry, sine gratings, etc.). However, it remains unknown if and how much visual patterns produce recognition probabilities equal to semantically complex, environmentally and/or socially valid stimuli. To address these questions, we conducted an experiment in which a human face expressing various emotional states was demonstrated during saccades; the participants were then required to identify the test stimuli in a two-alternative forced choice task. In addition, we assessed the observer’s accuracy of face localization in the visual field.

## 2. Materials and Methods

### 2.1. Apparatus

The experiments were carried out using a video-based eye tracker model iView X™ SMI Hi-Speed. Custom software was used for displaying stimulus images on and off the screen during saccadic eye movements [10]. Recording of subjects’ oculomotor activity was performed monocularly at a frequency of 1250 Hz. The display used was a color-calibrated ViewSonic G90fB CRT operating at 144 Hz refresh rate and 800 × 600 pixels resolution viewed from a distance of 57.3 cm with a chinrest. Screen angular size equaled 35.1 by 26.7 degrees.

### 2.2. Photo Database

Photos of a male face with explicit expressions of basic emotions: fear, anger, disgust, joy, sadness, surprise, and a neutral facial expression [11] (Figure 1). The database was collected earlier with ecologically valid criterion in mind; the facial expressions were captured on a high frequency video of transitions between types of emotions. The images were shot in color and standardized on a Russian sample by means of selecting the most appropriate frames from the video sequences with highest emotional identification ranks. The angular dimensions of the stimulus were approximately 3.7 by 6 degrees. The mean brightness was 39.2 cd/sq.m.

### 2.3. Participants

Students of Moscow universities with normal or corrected to normal vision were used as participants. The experiment involved 18 people (11 male, 7 female) aged 18–20; mean 19 ± 0.6 yrs. All subjects gave their informed consent for inclusion before they participated in the study. The study was conducted in accordance with the Declaration of Helsinki, and the protocol was approved by the Ethics Committee of Institute of Experimental Psychology, Moscow State University of Psychology & Education (Project identification code: No15-4-19, date: 6 May 2019).

### 2.4. Procedure

In the beginning of each trial, the participant fixed their gaze on a black cross (spanning square 0.95°) in the center of the screen. After a three-second interval, the cross was replaced by a laterally displaced object, which by experimental design should have initiated a saccade. The very same black cross was used for this purpose. It was shown randomly either to the left or right off the center of the screen at a distance of 10°. After approximately 2 degrees passed by the eye from the saccade’s onset, the test object was displayed (photo of the face). The target image was displayed in one of the following positions: center of the screen (0° displacement), the middle between the center of the screen and the laterally displaced cross (±5°), or right on top of the cross (±10°) (see Figure 2).

The time span of the test object was either 1 or 2 screen frames, comprising 6.9 to 13.9 ms. Stimulus duration was double-checked by a photodiode attached to the screen over the black/white field area, sending real-time TTL signals into the eye tracker’s live data feed. Therefore, we do know which trials contain stimulus of which duration, and we have filtered out trials which either contain longer stimulus than these values, were too noisy to trigger a 10-degree saccade, or are suspicious to have a stimulus of indeterminable duration. Participants were required to complete 100 trials, although most of them did not succeed in doing so either due to gaze tracking quality degradation over time, even after attempted recalibration, or fatigue. They identified the emotional modality of facial expression displayed during saccadic eye movements after 100 ms upon image removal. This hard-coded time delay was chosen so that it does not mask the visual memory of subjects, and yet it is barely noticeable or frustrating. They did so by pointing and clicking the image that seemed most probable to match the target stimulus out of the two alternatives, one of which always matched the target image, and another one selected randomly, and both presented next to each other at randomized order to prevent possible choice bias. The two alternative images were displayed in the center of the screen slightly above the fixation area, also to prevent masking. The participants were also asked to indicate the localization of the target face image by moving the rectangular frame, displayed on the monitor screen immediately after they made the choice of a facial expression.

### 2.5. Data Validity

After the initial data cleaning, 215 trials (13% out of 1699) were subject to statistical analysis. Characteristics of analysed saccades are shown in Table 1. Detection of oculomotor events (fixations, saccades, blinks) was provided by the I–VT algorithm (SMI IDF Event Detector 3.0.18 software utility) with the following parameters: saccade speed minimum threshold, 70 deg/s, and minimum fixation duration, 40 ms [12]. Data processing was conducted in R 3.5.1. Holm correction was applied for multiple comparison.

Very strict validity criteria were applied to filter out those trials in which either a saccade did not happen (i.e., subject blinked), or data feed was too noisy, or stimulus failed to disappear before the eye reached the target cross’ former position (mostly due to ‘unlucky’ display vertical scan phase) (see Table 2; Figure 3). It is worth noting that due to our software not utilizing VSync capabilities, the stimulus image can appear 1 frame earlier than the photodiode sends the TTL signal, but never later (black/white field was located right above the image position at the edge of the screen). Therefore, we have even stricter criterion which actually filters out more trials than needed. Even more than that, we’ve assumed the screen phosphorus noticeable decay time to comprise one extra frame (gray-to-gray), leading to all calculated stimuli durations being enlarged, some by more than 6.9 msec. Concluding all that, no valid trials contained stimulus images still being displayed while the gaze reached final position nor decelerated enough to say the saccade had already finished (nor had post-saccadic oscillation begun, see Figure 6, bottom).

## 3. Results

### 3.1. The Accuracy of Modality Identification in Facial Expression Stimuli

Accuracy levels were subject to nonparametric statistic tests. Kolmogorov–Smirnov test was conducted and normal distribution hypothesis refused. The average probability of correct identification of facial expressions during saccades for all experimental conditions was found to be above chance and equals 0.62 (Pearson chi square (1) = 11.63; *p* = 0.0006; 95% confidence interval (CI95) = 0.55–0.68). The correlation between the frequency of adequate recognition and the expression modality is also statistically significant (Pearson chi square (6) = 23.19; *p* = 0.0007). Expressions of joy (0.86), disgust (0.73) and fear (0.68) are most accurately recognized (Figure 4).

Generally, statistically significant predictors of correct emotion identification are the alternatives of neutral expression (0.71), sadness (0.69) and anger (0.64). The masking influence on the choice of target emotions shows up in cases with disgust (0.46) and joy (0.56). There is an inverse correlation of accuracy and alternative expression (Pearson r(5) = −0.76; *p* < 0.05). Most of incorrect responses were given in trials with neutral face stimulus (0.33), which at the same time has the smallest masking effect compared to basic emotions. On the contrary, the lowest inaccuracy rate is given in trials with facial expression of joy (0.86), which has a prominent masking effect (0.56) on the choice of other basic emotions, second to disgust (see Table 3).

Joy is not subject for biased effect, as shown by insignificant differences of accuracy between all types of given alternative choice. However, it does have a prominent advantage for correct identification in given conditions compared to neutral expression. We tend to attribute it with more diverse and protruding graphical features of the image itself (on the optical level), which gets interpreted as more socially valuable or important (on social level of perception and communication), and therefore gets more resources during cognitive analysis stage.

The generalized structure of erroneous responses in perception of facial expressions during saccades is shown in chord diagram (Figure 5). Arrows point at the emotions that are most often chosen in response to the displayed stimuli. The set of responses caused by a certain alternative expression characterizes its categorical field [13,14]. Notice that joy masks four other expressions, while anger and neutral, on the contrary, are the least mistaken.

### 3.2. Perceived Localization of Face Images

The assessment of stimulus localization during saccade differs from its actual position in all the locations of the test object in the visual field (Table 4). As can be seen from Figure 6, contrary to the actual position, the face image is seen by the observers in a relatively narrow area of visual half-field, close to the saccade end point.

The linear dependence of the perceived localization and actual localization of the test object is preserved within this area. Figure 6 (top) illustrates obtained probability densities of the test object localization. Perceived localization curves have peaks in the area of 9 degrees, but are also somewhat distributed along the stimulus exposition trace (Figure 6, bottom).

## 4. Discussion

The main results of the study show that adequate perception of facial expressions in a virtual communicant are found during rapid eye movements. The frequency of correct expression identifications at an average saccade velocity of about 189 deg/s equals 0.62. This is significantly higher than the frequency of detecting points of light or geometric shapes under similar conditions [1] and virtually coincides with identification of regular spatial patterns comprised of mathematical symbols [4]. In comparison with the results on perceiving facial expressions in condition of unconstrained examination, the obtained value is lower by 15–20%. According to data from earlier studies, the average frequency of adequate assessments of basic expressions by observers from various socio-cultural groups is 0.79 [15,16]. Introducing constraints (short exposure time, changes in egocentric orientation of the face or its elements, usage of noise masks, etc.) leads to degradation of recognition accuracy of same facial expressions. For instance, the accuracy value for a 3 s exposure of normally oriented face image is 0.92, but 0.67 for rotated downwards; for subtle or mixed emotional expressions, the values are even lower: 0.51 and 0.32, respectively [16].

Although the duration of one visual fixation (250–350 ms) is often sufficient to determine the modality of emotion [13], with short and ultrashort exposure times (*t* < 100 ms) the accuracy of expression recognition falls as low as 0.15–0.24 [17,18].

In fact, the average frequency of recognizing basic emotional expressions varies across a broad range of values depending on the conditions of the study and can be either higher or lower than the values obtained during saccadic eye movements. This means that accuracy of facial expression evaluations during saccades is determined by emotion modality rather than by hypothetical concept of “saccadic suppression”. Emotions of joy (0.86) and disgust (0.73) are best recognized, whereas anger (0.52) and neutral (0.33) expression are not recognized. Comparing these values with the results obtained in other studies on assessment of basic emotional expressions, it is easy to conclude that the only emotion that is most accurately and unambiguously recognized under various conditions is joy [14,19]. In cross-cultural studies [15,18] the average frequency of its identification is 0.95. Anger is poorly recognized, especially by the Japanese (0.56), which is in agreement with our results. Cross-cultural estimates of fear (0.74) virtually coincide with the measurements during eye movement (0.68). With further reduction of stimulus duration to 100 ms, efficiency of recognition of all basic expressions deteriorates significantly due to the weakening of configuration links [13].

The choice of a suitable emotion, which the participants make using two alternative images, depends on the modality of expression displayed during a saccade. The attractiveness of facial expressions and their visual salience plays a key role: if the image of an emotional expression stands out among others, it will facilitate the choice. However, if such an expression is presented as an alternative, it obscures the correct choice, inhibiting possibly correct response. We found that the neutral facial expression has the least negative impact on the test object recognition. Expressions of joy, fear and surprise, on the contrary, mask the target images. The higher the basic emotion’s recognition rate, the more effectively it masks other emotions. The neutral state is dualistic by its nature. On one hand, it is devoid of mimic signs of emotions; on the other, it contains potentially expressive features [13,20]. Therefore, in cases when a neutral expression is suggested as alternative response, it acts as a reference point and facilitates the choice of basic expressions. In those cases when the neutral expression becomes the test object, it reveals similarities with most of the basic expressions, making it difficult to identify the stimulus. The special status of the neutral state is manifested in the structure of erroneous perception of basic expressions. It is worth noting that just as in other experimental conditions, responses which do not coincide with the target expression can be only considered as relative mistakes. Seemingly incorrect choices are natural and reflect the ambiguity of facial expressions. Each facial expression is perceived as similar to a number of emotions, that is, it belongs not in a single perceptual *category*, but a constellation, or field. “Adequate perception” and its relation to accuracy of emotional recognition (identification) is only the core of the categorical field of this particular emotional state. According to the structure of wrong responses, categorical fields of basic expressions displayed during rapid eye movements largely coincide with those in more common conditions (during fixation and/or examination [13,21]. For instance, joy and surprise, and surprise and disgust are commonly confused with each other. A more general regularity may also be noticed: There is an inverse dependence of emotional recognition accuracy on the number of its neighbours in same category (represented by chords in the structure of erroneous perception).

Location parameters of correct response frequencies during the image exposure in different parts of right and left visual half-fields are not statistically different. First, this means that, like in normal conditions (with gaze fixations), the area of effective facial perception is not restricted by the size of the fovea centralis, but extends to nearby peripheral field. Second, perception of facial expressions can begin not only during gaze fixation, but also at peak speed of saccades (around 300–400 deg/s; see Figure 7), both in center and on the periphery of visual field. The spatiotemporal dynamics of the perceptual process are not being destroyed by the rhythmic structure (fixation–saccade–fixation) of the oculomotor activity, but are accompanied by it. A significant decrease (0.90 vs 0.73) of perisaccadic perception accuracy (but still above chance) was also shown by means of intra-saccadic surface changes of the background which yielded decrease in post-saccadic identification accuracy of letters [22]. In the context of gathered data, the notion of “saccadic suppression” as a mechanism regulating the perception of environmentally and/or socially valid objects appears to be questionable. The discrepancy between the evaluation results of simple and complex (environmentally/socially significant) stimuli shown in our experiment could be viewed as a procedural flaw: it was not possible to visually control declared parameters of color stimulus presentation during a given moment of saccade. Nevertheless, the temporal resolution criteria for acquired data is proven both by technological [10,23] and psychological evidence. By the latter we mean the transformation effects on the visual space–time, which are regularly reproduced during rapid eye movements, regardless of the stimulus type. These are the wrongly localized stimuli [24,25] and compression of their positions towards the saccade endpoint [26,27]. Both effects were confirmed during our experiment.

According to our data, regardless of the image location during the saccade, observers localize it in same narrow areas of the visual field nearby the forthcoming fixation point. The actual location of test images in regards to anticipated target remains unchanged during the trial, whilst subjectively perceived distance between them is significantly narrowed. The amount of such compression effect corresponds to the values described in earlier studies.

The experiment addresses the problem of visible world stability (constancy of visual direction): the relative independence of space–time state of perceived object with head and eye movements (locomotions) of the observer [28]. We have shown that disturbances of facial images’ perception stability during saccades occur very rarely and are likely associated with fatigue or distraction of participants. Despite the fact that the face projection moves upon the retina surface by several degrees (this value exceeds the threshold values of motion detection by two orders of magnitude), the perceived direction of the face remains the same, including the cases of mislocalization. Compression of perceived space does not affect the structure or stability of the complex object. In our experimental conditions, stability remains intact not only during eye movements as a whole, but also in separate parts of saccade trajectory, regardless of the time and location of its occurrence, either on acceleration, peak or deceleration phase (in the beginning, the middle or at the end of a saccade). Since these parts come into place spontaneously, unwillingly, and at very short timespans, describing partially stable perception of an environmentally significant trait either in terms of retinal–extraretinal signal interaction [29] or separation of optical invariants from the stimulus stream [30,31,32] appears to be highly problematic. All the data acquired in the present study suggests that while saccades are being performed, the visual channel remains available, capable of receiving and processing environmentally significant information at any time. Moreover, it is sensitive to the magnitude of misalignment between the gaze direction and perceived direction of the complex object, and, therefore, can be used for controlling the oculomotor act. Given the systemic nature of the visual process, it is assumed that control of saccadic eye movements is associated with the early and middle stages of perceptiogenesis in regards to facial expressions, and target-directed saccades are crucial to the development of visual sensation.

## 5. Conclusions

Complex environmentally or socially significant objects that are targets of attention during saccades are perceived by the observer differently than light dots, stripes or gratings. The probability of recognizing a facial expression—one of the most important stimuli in social behavior—is above chance during rapid eye movements and depends on the emotional modality. The identification accuracy, its predictors and characteristics of “mistakes” correspond to values of basic facial expression recognition shown in standard experimental conditions. The empirical data obtained in this study indicates the continuity of the visual process and possibility of its direct inclusion in the regulation of elementary eye movements [33,34]. The eccentric exposure of environmentally significant objects during saccades produces an increase in recognition efficiency compared with the reference level, and can possibly yield visual feedback. The rhythmic structure of the oculomotor activity does not break stages (phases) of the perceptual process, but binds them. Perception of facial expressions could start and continue not only during the period of stable fixation, but also at the peak speed of rapid eye movements.

## Figures and Tables

**Figure 1 behavsci-09-00131-f001:**
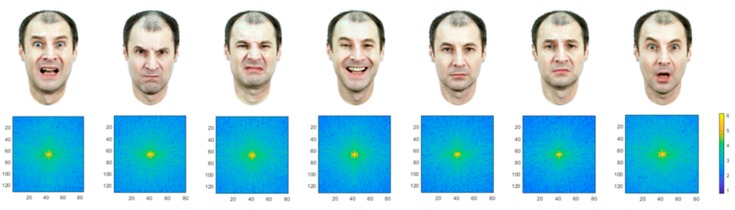
Stimuli images presented intrasaccadically along with their respective Fourier transforms. Left to right, top row: fear, anger, disgust, happy (joy), neutral, sadness, surprise. All images were presented on a 75% gray background.

**Figure 2 behavsci-09-00131-f002:**
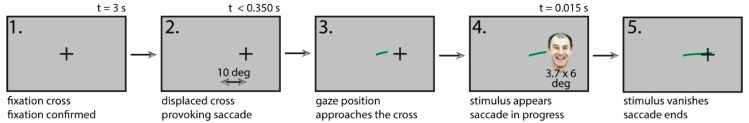
Experimental setup. 1. Subject fixates on the cross. The trial does not start until fixation is detected by software. 2. Fixation cross is displaced instantaneously by 10 degrees either left or right, in order to act as a saccade target (subjects are instructed to follow the cross as fast as they can). 3. While saccade is still running, software detects a simple gaze position displacement of at least 2 degrees off the screen center. 4. Stimulus image is presented on screen, in three possible positions (center, 5 degrees, or 10 degrees either left or right) for 1 or 2 display frames (which is proof-checked by photodiode). 5. By the moment the subject manages to reach the cross, the stimulus has already vanished.

**Figure 3 behavsci-09-00131-f003:**
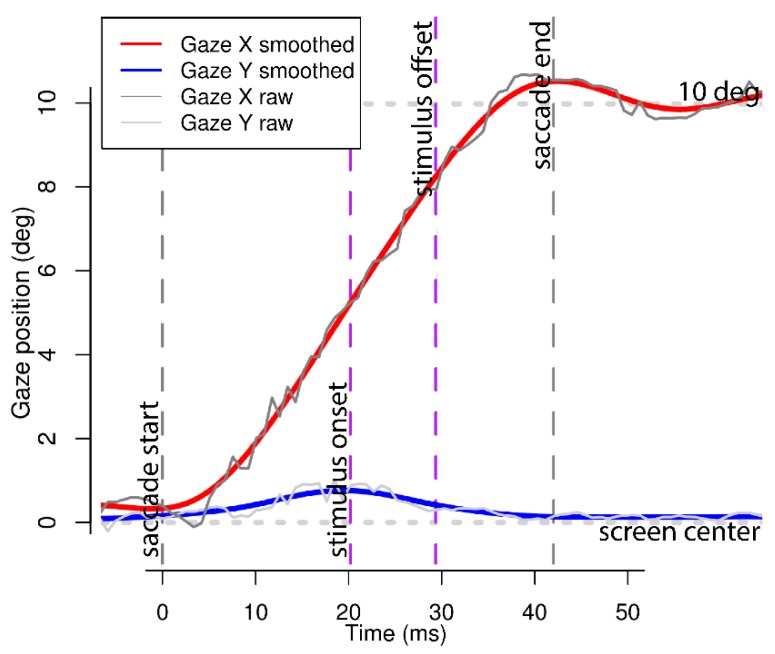
Typical gaze trajectory of a valid trial. Notice that only raw gaze data was subject to validity criteria and all statistical analysis.

**Figure 4 behavsci-09-00131-f004:**
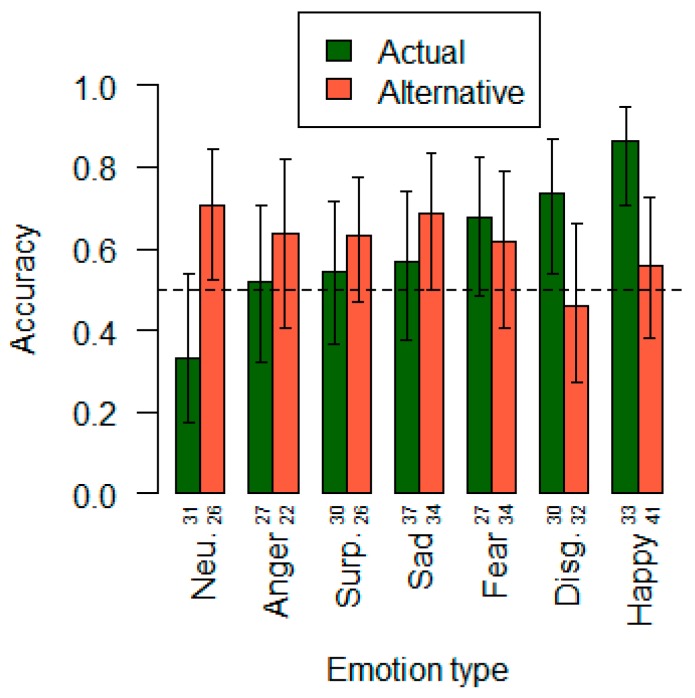
Accuracy bar plot by actual stimulus/alternative choice. Numbers beneath the bars indicate absolute frequencies of each condition in the sample. Vertical bars indicate 95% confidence intervals. Baseline is 0.5.

**Figure 5 behavsci-09-00131-f005:**
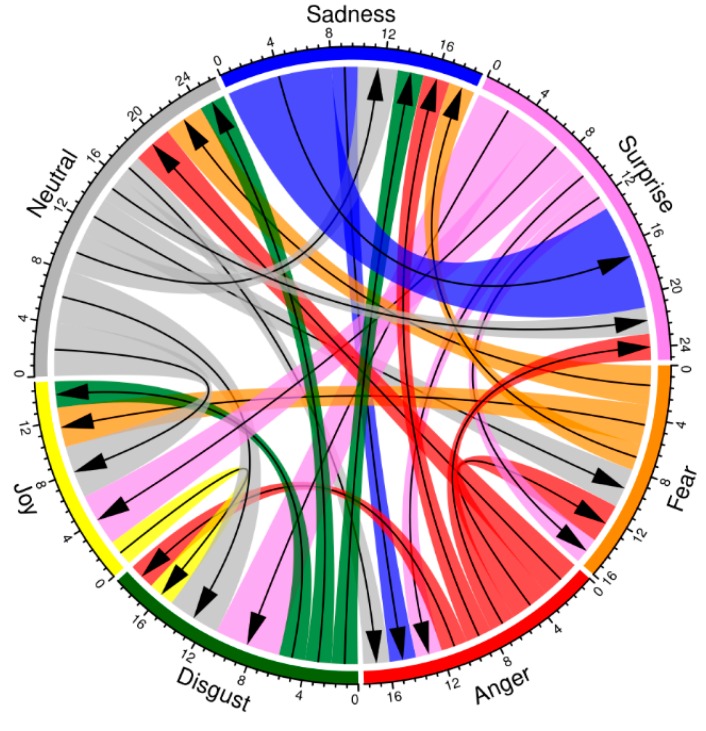
The structure of erroneous perception of basic emotional expressions during saccades. Arrows point at facial expressions which are preferred over the source. Radial axis scale represents absolute frequencies of erroneous responses. The least frequent are not shown for clarity.

**Figure 6 behavsci-09-00131-f006:**
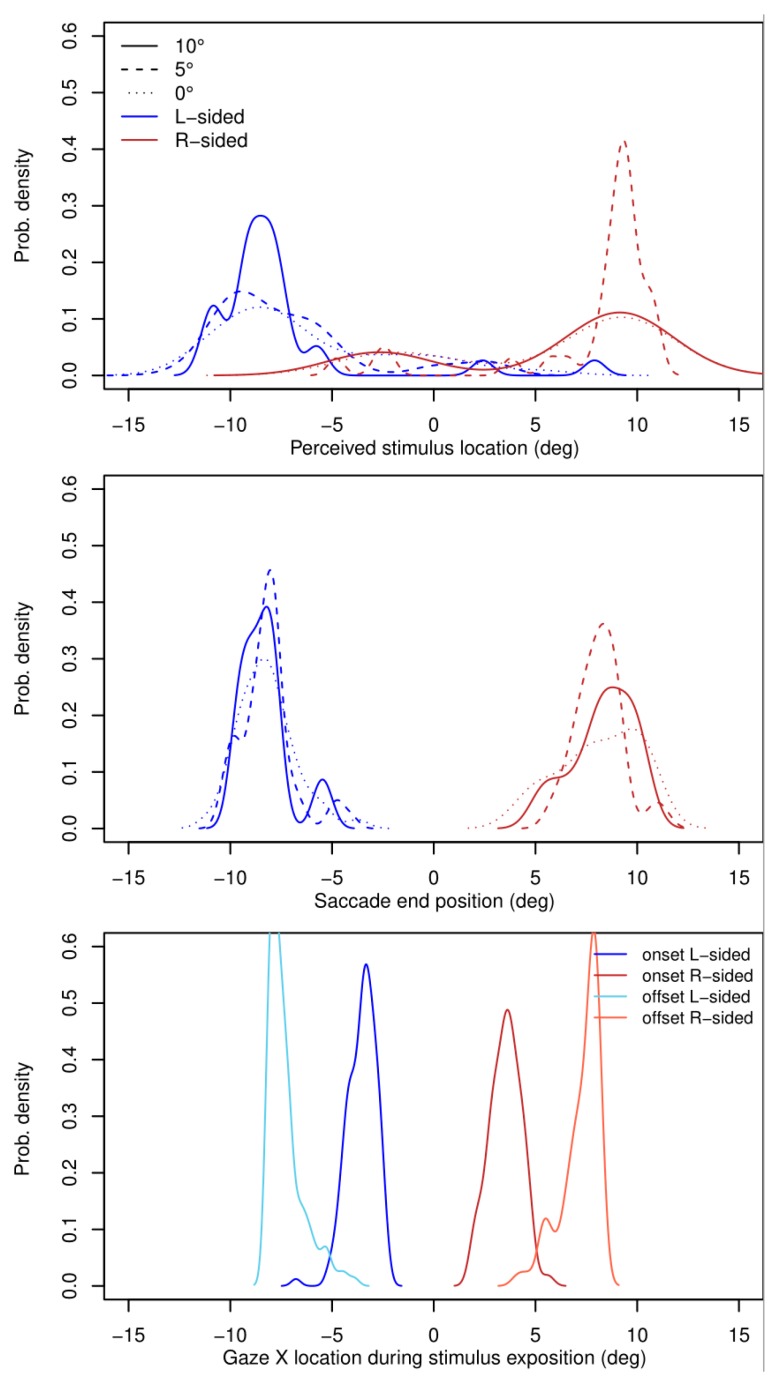
Probability density functions. (**Top**) Stimulus location as reported by naïve subjects. (**Middle**) Saccade endpoints. (**Bottom**) Gaze position at the moment of stimulus onset and offset. Each PDF is split by experimental condition and saccade direction. Deg = degrees.

**Figure 7 behavsci-09-00131-f007:**
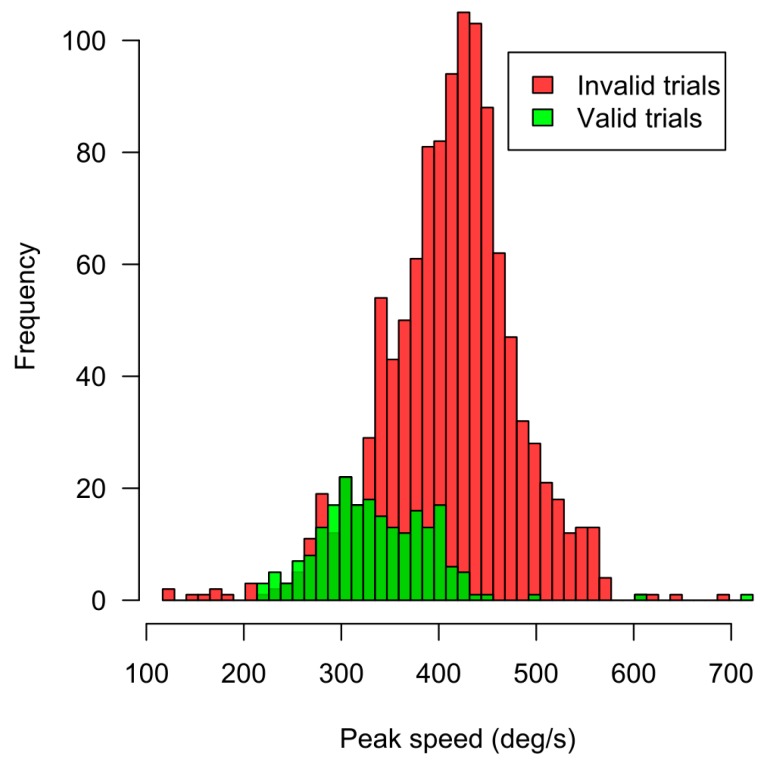
Saccade peak speed histograms in this study both for valid and filtered out trials.

**Table 1 behavsci-09-00131-t001:** Descriptive statistics of saccades from valid trials.

	Min	Max	Mean	SD
Saccade latency L-sided (ms)	86	349	200	63
R-sided (ms)	92	334	194	66
male (ms)	86	349	182 ^1^	61
female (ms)	105	337	223 ^1^	59
Saccade duration L-sided (ms)	32	75	45	8
R-sided (ms)	30	65	44	7
male (ms)	30	75	45	8
female (ms)	32	65	43	7
Saccade amplitude L-sided (deg)	4.5	11.1	8.3	1.2
R-sided (deg)	4.9	14.7	8.1	1.6
male (deg)	4.6	11.1	8.3	1.2
female (deg)	4.5	14.7	8.1	1.5
Peak saccade speed L-sided (deg/s)	214.4	602.7	335.7	59.5
R-sided (deg/s)	219.7	722.2	338.7	65.0
male (deg/s)	218.6	494.0	338.7	48.2
female (deg/s)	214.4	722.2	333.8	77.6
Average saccade speed L-sided (deg/s)	87.4	294.7	190.2	35.0
R-sided (deg/s)	94.6	234.8	187.4	30.4
male (deg/s)	87.4	237.9	188.2	33.0
female (deg/s)	94.6	294.7	190.7	34.3

^1^ Mann-Whitney U = 3327; *p* < 0.001; delta ≈ −45; CI95 ≈ −62–−27 (ms). SD = standard deviation.

**Table 2 behavsci-09-00131-t002:** Criteria for filtering invalid trials.

	Min	Max	% of Trials Filtered out
Saccade latency (ms)	75		20
Saccade latency (ms)		350	33
Gaze X position at stimulus onset (deg)	|1.85|		10
offset (deg)		|8.15|	66
Stimulus duration (ms)	6.9 ^1^		2
Stimulus duration (ms)		15.8 ^2^	15
Trial invalid due to interruption for recalibration			1

^1^ Durations reported by photodiode lower than this mean the white field flashed soon after the trial began (during the running frame refresh cycle), most often due to noisy gaze detection at the moment which triggered a false saccade detection. ^2^ Durations higher than this mean stimulus failed to disappear after 2 frames, due to software timing being imperfect. Notice that sensor timings are not exactly a multiple of frame duration.

**Table 3 behavsci-09-00131-t003:** *p*-values for pairwise comparison of accuracy grouped by stimulus types, chi square with Holm correction.

		Actual Stimulus Image					
		Fear	Anger	Disgust	Joy	Neutral	Sadness
**Alternative choice image**	Anger	1.0					
	Disgust	1.0	1.0				
	Happy	1.0	0.11	1.0			
	Neutral	0.30	1.0	0.11	0.0008		
	Sadness	1.0	1.0	1.0	0.24	1.0	
	Surprise	1.0	1.0	1.0	0.13	1.0	1.0

**Table 4 behavsci-09-00131-t004:** Actual and perceived localization of the stimulus, estimate for Wilcoxon W.

Actual Position	Perceived
10 degrees left (deg)	−8.5
Right (deg)	+7.9
5 degrees left (deg)	−7.8
Right (deg)	+9.1
Center with left-displaced target cross (deg)	−6.7
Center with right-displaced target cross (deg)	+5.7

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
