# Peer review of "Visual Perception of Facial Emotional Expressions during Saccades"

_behavsci, 2019, doi:10.3390/bs9120131_

Round 1
Reviewer 1 Report
This experiment demonstrates above-chance levels of facial emotion discrimination in stimuli presented while the eyes are moving during a saccade. This result is used to argue against the long-standing concept of saccadic suppression.
There are two aspects of these results that are not explored much. The manuscript argues that accuracy in this emotion discrimination task is higher than in much simpler discrimination tasks with abstract stimuli. However, there is not much discussion of why identification of simple stimulus properties should be so heavily suppressed during saccades if facial emotion identification is not. There is certainly precedent for accuracy being higher for complex discriminations and lower for simple discriminations. For instance, Fei-Fei Li showed that while subjects could detect the presence of animals or vehicles in photos at above-chance levels in unattended regions of a display, they performed much worse at discriminating the relative locations of red and green regions in a simple circle (Li, VanRullen, Koch, & Perona, PNAS, 2002, 99(14), 9596-9601). In the saccade task presented here, is there more that could be said about why emotion discrimination might be more accurate than more simple discrimination tasks?
Also, it is very interesting that when subjects report the location of the face stimulus that appears during a saccade, they tend to report locations near the target of the saccade, even for stimuli that appear nearer to the starting point of the saccade. What does this misperceived location tell us about the processing of information during a saccade?
If I understand correctly, new face stimuli were created specifically for this experiment. I would help to be able to see those stimuli.
The discussion section makes general claims about how the different emotional expressions differ from one another in their discriminability. However, if I understand correctly, this experiment was done with just a small number of photos of a single individual. The discussion section does point to similar results from other studies, but based on the limited stimulus set here, it is difficult to make general conclusions about the relative discriminability of different emotions. At the very least, there should be a warning to readers that these conclusions are limited by the small stimulus set.
What sort of display monitor was used to present the face stimuli? To what extent was it possible for the software to determine the exact exposure time of the stimulus, and how much uncertainty in the onset and offset of the face stimulus was introduced by the monitor?
What does it mean that the face stimuli were standardized on a Russian sample (line 55)?
I did not find an explanation of the different line types used in the graph in Figure 2. Perhaps they are similar to the line types used in Figure 5b.
It takes some work to understand the presentation of the results in the text and the graphs. It looks like both commas and periods are used as decimal markers, which adds to the confusion.
What is the definition of “adequate recognition” in this manuscript?
For Figure 4, it is unclear exactly what criteria were used to determine when an arrow is drawn, and when a bold arrow is drawn. It would also help to have a fuller explanation of what is symbolized by the direction of the arrows.
Many parts of the manuscript are difficult to understand. Here are a few examples.
Lines 246-248: “the reliability criteria for acquired data do exist and lie both in the technical and in the psychological sphere of knowledge”.
The last paragraph of the discussion section (lines 258-279) is difficult to follow.
Line 290: “can act visual feedback”
In summary, these results have potentially important implications for understanding the nature of saccadic suppression, and the location errors and the higher accuracy for emotion discrimination relative to simpler discrimination tasks may be signaling something important about other aspects of visual perception as well. However, aspects of the text, the presentation of data, and the graphs, could be clarified, and some of the conclusions are perhaps broader than is warranted by the results.
Reviewer 2 Report
The authors report an experiment in which participants could discriminate facial expressions shown to them only during the rapid saccadic eye movements.
It is commonly assumed that visual information uptake is largely suppressed during saccades, which is why vision is often viewed as a discrete process, in the sense that it is confined to periods of eye fixation. There is only little data showing that visual information is processed even when it is available only during the saccade (e.g., Uttal & Smith, 1968, Perception & Psychophysics; Balsdon et al., 2018, Consciousness and Cognition). Therefore, the present study extends the current understanding by turning to stimuli of facial expressions. The present results are very interesting to this line of research, because they provide converging evidence and introduce a new class of stimuli that may be more ecologically valid than the previous ones. However, there are a number of points that should be addressed before the manuscript is published. I detail my points below.
Major points:
1) The present experiment investigates stimulus presentation within the narrow time window of a saccadic eye movement. Therefore, to ensure that stimuli really did not extend beyond this time window, it has to be checked how long stimuli persisted on screen after their programmed offset. Stimulus persistence depends on the specific monitor. For gray-to-gray transitions, the on-screen-persistence can be in the range of several tens of milliseconds (e.g. Poth et al., 2018, Behavior Research Methods). Thus, for the specific stimuli used here, the authors should externally measure their persistence (the decay of the stimulus after its issued offset) and show that it fully dropped to baseline within the saccade. The methods for doing this (measurement circuit, etc.) can be found in one of our papers, for example (Poth et al., 2018, Behavior Research Methods).
2) The authors motivate their study by stating that their stimuli are more ecologically valid than the ones previously used (e.g. lines 37-39). From that I would have expected a comparison of their more naturalistic stimuli with more traditional simple stimuli. Indeed, I think it would be highly interesting to conduct an experiment where these stimulus classes are compared regarding their intrasaccadic perception.
3) Procedure: Were the locations of the two alternative choice stimuli randomized across trials (with each location occurring equally often)? Such a two alternative forced choice task can enable to analyze performance in terms of d’ rather than the proportion of correct responses, which controls for participants’ response biases.
4) Results: Related to the previous point, I suggest that the authors compute d’ for each participant in each condition and then analyze these data with the statistical tests appropriate for the data (e.g. ANOVAs, t-tests or Wilcoxon-tests, etc.).
5) Results, line 101: Why did female participants exhibit a higher saccade latency?
6) Results: Pairwise comparisons should be provided to compare the different facial expressions.
7) Discussion: The authors might want to reflect their findings more in the light of studies assuming that visual perception across saccades happens in a discrete fashion (e.g., Hollingworth et al., 2008, Journal of Experimental Psychology: General; Tas et al., 2012; Journal of Vision; Poth et al., 2015, Frontiers in Systems Neuroscience; Poth & Schneider, 2016, Journal of Vision). Specifically, these studies assume correspondence mechanisms bridging the input interruption caused by saccades, so that I am curious about what the authors think of these mechanisms when considering their findings of intrasaccadic perception.
8) Discussion: The authors should also discuss the spatial frequencies of their stimuli, because these may affect how stimuli can be processed while the retina is moving.
Minor points:
1) Related to major point 1, the authors should report their monitor.
2) Introduction, lines 35-39: The statements should be supported by references in the appropriate places.
Reviewer 3 Report
Barabanshikov and Zherdev presented an interesting paper that showed that when participants “see” a face depicting an emotion (target) during a saccades, then the probability of choosing the image that seemed more similar to the target stimulus out of two alternatives is higher than chance.
The paper is well written and very interesting, but I can’t recommend publication in its current form.
I have some issues that needs to be addressed.
Major
The frequency of correct test-object identification is higher with joy. Could you please explain if you need (or not and why) a control study to check whether there is a general bias to select joy and if this bias can influence or not your results? For example, if a participant have a bias in selecting the joy emotion he would have many hits (selecting joy when it was shown a face depicting a joy emotion) but also many false alarm (selecting joy when it was shown a face depicting a different emotion) when the correct response is a different emotion. Were this type of error considered in your statistic? (In fig 3b, you present the correct rejection [if I understand correctly the figure], reject joy emotion when the target is not joy emotion, maybe this can help you). The Pearson chi square is a statistical test applied to sets of categorical data to evaluate how likely it is that any observed difference between the sets arose by chance. But it does not tell you if a particular set has more observation than the others. I do not understand how you found that you have better performance (statistically) with joy compared to other emotions as your Fig. 3a showed (***).
Am I Wrong? Could you please explain better the figures and the result section?
This manuscript does not consider an important aspect that could add importance to this paper. In the literature is reported that different emotion are better elaborated when specific spatial frequencies are available.
For example: Vuilleumier, P., Armony, J. L., Driver, J., & Dolan, R. J. (2003). Distinct spatial frequency sensitivities for processing faces and emotional expressions. Nature neuroscience, 6(6), 624.
What happens during the saccades? Are only the low spatial frequencies available?
(Garcı́a-Pérez, M. A., & Peli, E. (2001). Intrasaccadic perception. Journal of Neuroscience, 21(18), 7313-7322.)
Does the better performance observed with joy emotion could be explained considering that one can elaborate properly the joy emotion even without the high spatial frequencies?
Minor point
Lines 77, please check this line, maybe you left a double space after the dot. Lines 121, 126-127. I think a dot is needed instead of a coma. Lines 120-121. I would add some word to explain better figure 3b and the analysis that you have conducted.
Suggestion
You used non-parametric statistic. I would say why (data are not normally distribuited etc.) I think that there is a version of kruskal wallis test (Scheirer-Ray-Hare extension) that can be considered a sort of “non parametric anova”. I am not sure about that. Perhaps the authors can increase the importance of the paper with this additional analysis. But as I said I am not sure if you can use that test with your data. I read some papers that report that there is a curvature in the saccades when a social cue was elaborated. Maybe the following paper can be interesting for you (does your saccades curvature is different when the emotion is correctly recognize compared to a neutral condition or compared to a saccades executed during an incorrect trial where the emotional elaboration (likely) failed?)
In case these papers are not relevant please ignore this suggestion
-Dalmaso, M., Castelli, L., Scatturin, P., & Galfano, G. (2017). Trajectories of social vision: Eye contact increases saccadic curvature. Visual Cognition, 25(1-3), 358-365.
Kompatsiari, K., Ciardo, F., Tikhanoff, V., Metta, G., & Wykowska, A. (2018). On the role of eye contact in gaze cueing. Scientific reports, 8(1), 17842.
Dalmaso, M., Galfano, G., & Castelli, L. (2015). The impact of same-and other-race gaze distractors on the control of saccadic eye movements. Perception, 44(8-9), 1020-1028.
Dalmaso, M., Edwards, S. G., & Bayliss, A. P. (2016). Re-encountering individuals who previously engaged in joint gaze modulates subsequent gaze cueing. Journal of Experimental Psychology: Learning, Memory, and Cognition, 42(2), 271.
Dalmaso, M., Castelli, L., & Galfano, G. (2017). Attention holding elicited by direct-gaze faces is reflected in saccadic peak velocity. Experimental Brain Research, 235, 3319-3332
Round 2
Reviewer 2 Report
The authors addressed all comments, so that the manuscript seems ready for publication.